# The Middle Ear Microbiota in Healthy Dogs Is Similar to That of the External Ear Canal

**DOI:** 10.3390/vetsci10030216

**Published:** 2023-03-11

**Authors:** Caroline Leonard, Pierre P. Picavet, Jacques Fontaine, Cécile Clercx, Bernard Taminiau, Georges Daube, Stéphanie Claeys

**Affiliations:** 1Department of Clinical Sciences, Faculty of Veterinary Medicine, B67 Sart Tilman, University of Liege, 4000 Liege, Belgiumcclercx@uliege.be (C.C.);; 2Fundamental and Applied Research for Animal & Health (FARAH), Food Science Department, Faculty of Veterinary Medicine, B43b Sart Tilman, University of Liege, 4000 Liege, Belgium

**Keywords:** canine, microbiology, veterinary medicine, next generation sequencing, tympanic bulla

## Abstract

**Simple Summary:**

Otitis media can be a consequence of chronic otitis externa and could represent a perpetuating factor. Sparse information is available concerning the normal microbiota of the middle ear. The objective is to compare the tympanic bulla (TB) with the external ear canal (EEC) microbiota in healthy dogs. Six healthy experimental Beagle dogs were selected based on the absence of otitis externa, negative cytology and bacterial culture from the TB. Samples from the EEC and TB were collected directly after death using a total ear canal ablation and lateral bulla osteotomy. The 16S rRNA gene amplicon was analyzed. No significant differences between the EEC and TB microbiota were noted for alpha and beta diversity. The microbiota profile was similar in the EEC and the TB of the Beagles.

**Abstract:**

Otitis media can be a consequence of chronic otitis externa and could represent a perpetuating factor. While the microbiota of the EEC in healthy dogs and in the presence of otitis externa has been described, only sparse information is available concerning the normal microbiota of the middle ear. The objective was to compare the tympanic bulla (TB) with the external ear canal (EEC) microbiota in healthy dogs. Six healthy experimental Beagle dogs were selected based on the absence of otitis externa, negative cytology and bacterial culture from the TB. Samples from the EEC and TB were collected directly after death using a total ear canal ablation and lateral bulla osteotomy. The hypervariable segment V1–V3 of the 16S rDNA was amplified and sequenced with a MiSeq Illumina. The sequences were analyzed by the Mothur software using the SILVA database. No significant differences between the EEC and TB microbiota for the Chao1 richness index (*p* = 0.6544), the Simpson evenness index (*p* = 0.4328) and the reciprocal Simpson alpha diversity (*p* = 0.4313) were noted (Kruskal-Wallis test). A significant difference (*p* = 0.009) for the Chao1 richness index between the right and left EEC was observed. The microbiota profile was similar in the EEC and the TB of the Beagles.

## 1. Introduction

Otitis media, defined as an inflammation of the middle ear (ME), represents a common complication in canine recurrent/chronic otitis externa (OE), with a prevalence varying between 52% and 82.6% [1,2]. Infections in the ME could be a consequence of the extension of exudate and infectious organisms from the external ear canal (EEC) through a defective tympanic membrane (eroded/ruptured), resulting from the production of different proteolytic enzymes (proteases, collagenases, elastase, lysozymes) by phagocytic cells [2]. Healing of the tympanic membrane is possible during this process, as shown by Cole et al., who reported that 70% of dogs with a concomitant otitis media had an intact eardrum [3]. Infection in the ME has also been identified as a primary process and, in such case, represents one of the perpetuating factors of the OE [3].

In chronic suppurative otitis, medical treatment is dependent on the EEC structural changes, the presence or absence of tympanic membrane, as well as the overgrowth of micro-organisms and their antibiotic sensitivities [4]. The presence of resistant bacteria represents a therapeutic challenge and is, to date, a great concern [4,5]. Moreover, the identified bacteria in the EEC could differ from the ones in the ME, leading to therapeutic failure. By improving the knowledges of the tympanic bulla (TB) bacterial flora composition, better success of the medical treatment could be expected, making it possible to avoid surgery (total ear canal ablation and lateral bulla osteotomy) [6].

Many studies using 16S rRNA gene amplicon analysis profiling have described the microbiota of the EEC in normal and atopic dogs, with or without OE [7,8,9,10,11,12,13]. These studies confirmed that normal EEC microbiota is much more complex than supposed with the bacterial cultures [7,8]. Regarding the ME bacterial population, few data are available and the results are based on bacterial cultures only [3,14,15,16,17,18,19,20].

The objective of this study was to identify the microbiota of canine TB in healthy experimental Beagle dogs and to compare it with the EEC of these same dogs.

## 2. Materials and Methods

### 2.1. Study Population

Twelve experimental healthy Beagle dogs, housed in the same kennel, were recruited based on the absence of a history of ear disease or skin problems. These dogs were euthanized, for a reason unrelated to this study, with an agreement of the parent ethical committee.

### 2.2. Sample Collection

Samples were collected within six to 12 h after death. The right and the left ears were sampled independently in all of the dogs. The EEC and TB from the same side in each dog were sampled directly one after the other.

The EEC was sampled using FLOQSwabs (Copan Diagnostic; Murrieta, CA, USA), by rubbing the skin of the vertical EEC for 5 s. The samples were stored at −80 °C until DNA extraction.

The TB was sampled during a total ear canal ablation and lateral bulla osteotomy (TECALBO) surgery under aseptic conditions [21]. After the ablation of the EEC, the TB was sampled with the same method used for the EEC and stored at −80 °C until DNA extraction. The TB was also sampled using eSwabs (Copan Diagnostic; Murrieta, CA, USA) for direct cytology and bacterial culture. For the bacterial culture, the samples were sent to an external laboratory within 12 h. The samples for the bacterial culture were stored at 4 °C before analysis. The bacterial culture was initiated on blood agar plates TSS (Trypcase Soy agar + 5% sheep blood, BioMerieux, Marcy l’Etoile, France), Columbia CAP selective agar (Becton Dickinson, GmbH, Heidelberg, Germany) and MacConckey agar (Merck KGaA, Darmstadt, Germany). Microbial identification was performed with a matrix assisted laser desorption ionization-time of flight mass spectrometry (MALDI-TOF MS) (Bruker Daltonics, Billerica, MA, USA).

To select healthy dogs, avoiding cases of non-clinical ear problems, dogs with a presence of bacteria and/or inflammatory cells at direct cytology and/or with a positive aerobic bacterial culture were excluded from this study. Indeed, some dogs may have had an ear infection with secondary bacterial/fungal proliferation without a history of ear problems (aural pruritus, head shaking, discharge and/or rubbing of the ear for example) mentioned by the owners or with mild signs that can be missed.

### 2.3. 16S rDNA Extraction and High Throughput Sequencing

The DNA extraction, amplification and sequencing were performed as previously described [22].

The total bacterial DNA was extracted from the swabs using the DNEasy Blood and Tissue kit (QIAGEN Benelux BV; Antwerp, Belgium) following the manufacturer’s recommendations. Negative control samples with sterile swabs were included. This protocol was preceded by a beat beating step with glass beads >106 μm and glass beads soda-lime (Sigma-Aldrich, Overijse, Belgium, Cat. G4649 and Z265926).

PCR-amplification of the V1–V3 hypervariable regions of the 16S rDNA and library preparation were performed with the following primers (with Illumina overhand adapters), forward (5′-GAGAGTTTGATYMTGGCTCAG-3′) and reverse (5′-ACCGCGGCTGCTGGCAC-3′). Each PCR product was purified with the Agencourt AMPure XP beads kit (Beckman Coulter; Pasadena, CA, USA) and submitted to a second PCR round for indexing, using the Nextera XT index primers 1 and 2. After purification, the PCR products were quantified using the Quant-IT PicoGreen (ThermoFisher Scientific; Waltham, MA, USA).

A final quantification, by quantitative (q)PCR, of each sample in the library was performed using the KAPA SYBR” FAST qPCR Kit (KapaBiosystems; Wilmington, MA, USA) before normalization, pooling and sequencing on a MiSeq sequencer using V3 reagents (Illumina; San Diego, CA, USA). A positive control, using the DNA from 20 defined bacterial species, and a negative control (from the PCR step) were included in the sequencing run [22].

### 2.4. Data Analysis

Alignment and clustering were performed with the MOTHUR software package (v1.41.0) using a clustering distance of 0.03. The taxonomical assignments were based upon the SILVA database (v1.32) (https://www.arb-silva.de, accessed on 6 January 2021) of the full-length 16S rDNA sequences [23]. The VSEARCH algorithm was used for chimera detection as previously described [22,24,25].

Good’s coverage estimator was used as a measure of the sampling effort for each sample.

The negative controls, as a measure of determining erroneous results due to contamination, were not sequenced as there was no detectable amplification product in the samples. Suspected contaminants found in the controls (such as chloroplasts) were removed by filtering them from the OTU table, as previously described [26].

Subsample datasets were obtained and used to evaluate the alpha diversity (richness estimation—Chao1 estimator, microbial biodiversity—reciprocal Simpson index and the population evenness—derived from Simpson index) using MOTHUR.

Beta-diversity (bacterial community composition) was assessed with MOTHUR using the distance matrix based on the Bray–Curtis dissimilarity index and phylogenetic distance matrix, using a weighted UniFrac that was evaluated with AMOVA (with 10,000 iterations). Beta-dispersion was assessed with HOMOVA (10,000 iterations) in MOTHUR. Ordination analysis and 3D plots were performed with the Vegan (https://CRAN.R-project.org/package=vegan, accessed on 6 January 2021), Vegan3d (https://CRAN.R-project.org/package=vegan3d, accessed on 6 January 2021) and rgl packages (https://CRAN.R-project.org/package=rgl, accessed on 6 January 2021) in R (R: A Language and Environment for Statistical Computing, R Foundation for Statistical Computing, Vienna, Austria, 2015; https://www.R-project.org/, accessed on 6 January 2021).

As most of the datasets did not meet the assumptions of normal distribution, nonparametric statistical tests were used. Both Bray-Curtis (taxonomic evaluation) and weighted UniFrac matrix (phylogenetic approach) were used to evaluate the microbiota differences between the two groups. Statistical differences of the alpha diversity and population abundance between the dog groups were assessed with the Kruskal–Wallis H test, corrected for multitesting (Dunn’s test) using PRISM 7 (Graphpad Software; San Diego, CA, USA); differences were considered significant for a *p*-value < 0.05. The statistical difference of the population abundance between the groups was assessed with the Kruskal–Wallis H test, corrected or not with Storey false discovery rate, and followed by the Tukey–Kramer post hoc test, using the STAMP software [27].

## 3. Results

### 3.1. Study Population

From a total of 12 experimental Beagle dogs, six were excluded from the study due to positive cytology and/or bacterial culture. Finally, both ears from six Beagle dogs were included, giving a total of 12 ears. The dogs were all two year old males.

### 3.2. Most Common Taxa

The mean value of the Good’s coverage index was 99.76%. From 8,201,002 raw reads, a total of 7,790,951 reads were obtained after the cleaning step, with a median read length of 526 nucleotides. Finally, 10,000 reads per sample were retained for the OTU clustering and taxonomic assignment.

A total of 24 phyla were identified for the EEC and the TB. For both of them, Firmicutes was the most common phylum, followed by Proteobacteria, Actinobacteria and Bacteroidetes.

For the EEC, the three most common genera were for Firmicutes: *Turicibacter* spp., *Lactobacillus* spp. and *Atopostipes* spp.; for Proteobacteria: *Escherichia-Shigella* spp., *Enhydrobacter* spp. and *Psychrobacter* spp.; for Actinobacteria: *Corynebacterium* spp., *Micrococcus* spp. and *Micrococcacae genus* spp.; and for Bacteroidetes: *Prevotella* spp., *Bacteroides* spp. and *Porphyromonas* spp. From the total genera, the most abundant genus was *Turicibacter* spp., followed by *Lactobacillus* spp., *Escherichia-Shigella* spp. and *Corynebacterium* spp.

For the TB, the three most common genera were for Firmicutes: *Lactobacillus* spp., *Turicibacter* spp. and *Jeotgalicoccus* spp.; for the Proteobacteria: *Escherichia*-*Shigella* spp., *Psychrobacter* spp. and *Haemophilus* spp.; for Actinobacteria: *Corynebacterium* spp., *Kocuria* spp. and *Actinomyces* spp.; and for Bacteroidetes: *Porphyromonas* spp., *Alloprevotella* spp. and *Prevotella* spp. From the total genera, the most abundant genus was *Lactobacillus* spp., followed by *Escherichia-Shigella* spp., *Turicibacter* spp. and *Jeotgalicoccus* spp. A high abundance of *Escherichia-Shigella* spp. was observed in the right TB in one dog (Beagle number 7). (Figure 1).

### 3.3. Richness, Evenness and α-Diversity

The Chao1 richness index (richness estimator), the reciprocal Simpson biodiversity index (biodiversity estimator) and the population evenness index (considering distribution of the OTUs between the groups) were not significantly different between the EEC and the TB of the healthy experimental Beagle dogs (*p* = 0.6544, 0.4313 and 0.4328, respectively). (Figure 2) However, a significant difference (*p* = 0.009) was observed between the right and the left EEC for the Chao1 richness index (richness estimator).

### 3.4. Beta-Diversity

The non-metric multidimensional scaling (NMDS) graph showed no significant clustering between the EEC and TB of the experimental Beagle dogs by using AMOVA analysis with the taxonomic approach (Bray-Curtis). (Figure 3).

## 4. Discussion

To the best of the author’s knowledge, this is the first evaluation of the microbiota profile of TB in healthy dogs, using the 16S rRNA gene amplicon sequencing method. The microbial population of the ME, based on a culture-based method, has only been evaluated in one study in healthy dogs [14] and in few studies in dogs with otitis externa [3,15,16,17,18,19,20]. Therefore, a comparison of the results using the two methods is not reliable. According to Hettlich et al. and Palmeiro et al. [16,18], the most common species highlighted in the ME of dogs with otitis (culture-based method), when a TECALBO is depicted, were *P. mirabilis*,* P. aeruginosa*,* Streptococcus* spp., *Enterococcus* spp. and *Staphylococcus* spp., which are the classical strains also found in otitis externa. These bacteria were generally not observed in the tympanic bulla of the healthy experimental Beagle dogs in our study.

The present study shows that the microbiota profiles of the EEC and the TB were similar in the healthy experimental Beagle dogs. This observation differs from what is known in human beings. In healthy adult patients, a significant difference was noted between the microbiota profiles of the ME, the EEC and nasopharynx [28]. In children with acute otitis externa, Brugger et al. reported that the ME microbiota is not necessarily reflected by the one of the nasopharynx [29]. Another theory is defended by Man et al., which suggests that the ME is seeded by the nasopharynx microbiota in an ascendent manner by the Eustachian tube, with a substantive qualitative and moderate quantitative correlation between the ME and nasopharynx [30]. The aetiologies and pathomechanisms of the chronic otitis media in human beings cannot be compared to those in dogs [31,32]. In contrast to acute otitis in dogs, acute otitis media in human beings is essentially of nasopharyngeal origin and considered as a consequence of a viral upper respiratory, leading to the accumulation of mucus secretions in the middle ear due to the dysfunction of the Eustachian tube drainage and secondary ascendant bacterial infection [28,33].

According to our results, the hypothesis of an occasional tympanic permeability could be proposed in dogs, explaining a descendant seeding of bacteria from the EEC to the ME. Indeed, for unknown reasons, the eardrum could become transiently permeable. This would allow the bacteria present in the EEC to migrate to the TB and seed it. This could explain the similarity of the bacterial populations between the two compartments. A significant difference between the right and the left side was observed. This observation could be explained by the hypothesis that some environmental factors could be taken into account in this process, for example, the air flow in the kennel, the layout of the kennel or the dog’s habits. These environmental factors, together with potential traumatic factors/mechanical insult, could also explain why, in clinical settings, unilateral otitis or more severe infection on one side compared to the other are observed in some dogs.

The major phyla observed in the experimental Beagle dogs (Firmicutes, Proteobacteria, Actinobacteria and Bacteroidetes) are consistent with the previous studies evaluating the EEC microbiota profiles in healthy dogs [7,8,9,11,12]. The prevalence of each phylum was different from one study to another. However, a strict comparison of the results cannot be conducted as the 16S rRNA gene amplicon sequencing techniques were not identical [8,9,11,12].

The high abundance of the genus *Lactobacillus* spp. and *Turicibacter* spp. observed in the EEC in the present study was not noted in the previous studies in healthy owned dogs, using the same technique in the same laboratory [7]. Such divergence might be explained by environmental conditions (housed in kennel versus home) but also by individual and genetic factors (breed), as mentioned in other studies [13,34,35].

The major limitations of this study were the small number of samples (12 EEC and TB coming from six Beagle dogs) and the use of experimental dogs living all together with a lack of racial, sex and age variabilities. On the other hand, the sampling in one breed and dogs of the same age and sex reduces a possible individual variation.

Another limitation is the time at which the samples were obtained (within 6 to 12 h after death). Data concerning the evolution of bacterial population immediately after death are lacking in the literature. The studies mainly report the evolution of the bacterial community several days or weeks after death as this information can be useful to the medical examiners. This phenomenon (referred to as thanatomicrobiome), as well as possible contamination by environmental colonization (referred as necrobiome), could have influenced our results [36]. Another phenomenon might have affected our results; namely, bacterial translocation from one organ to another, which can be observed within 5 min of death [37]. Translocation is a function of the type of bacteria; for example, it has been demonstrated that *Proteus* spp. (not a major phylum in our study) can easily migrate from one organ to another [38]. Storage at a low temperature (4 °C) is not a guarantee of stopping the process because certain bacteria develop more at 4 °C than at 20 °C [39]. However, this phenomenon is likely important in organs where the density of the bacterial population is high, such as in the gut, but probably of much less concern in ears. Indeed, the density of the bacterial population in the two compartments sampled was low (with negative bacterial cultures). Furthermore, the aim of this study was to compare two compartments (EEC and TB), and both were assessed at the same time and in the same conditions.

The comparison with the other studies on the TB flora is limited given that the samples were taken by myringotomy in one other study, in healthy dogs (with a culture-based method); a contamination by passage in the external ear canal exists and is debatable [20]. Aseptic TECLABO is a way to avoid the contamination of the external ear canal.

Finally, a comparison of our results from the tympanic bulla with samples from the Eustachian tube would have been interesting.

## 5. Conclusions

The present study shows a similar microbiota profile in the EEC and TB in healthy experimental Beagle dogs. The microbiota profiles of the EEC and TB in the healthy dogs could be dependent on environmental factors. Although the microbiota profiles of the EEC and the TB in healthy dogs could be dependent on environmental factors, the presented data are important for further studies on the TB microbiota involved in OM.

## Figures and Tables

**Figure 1 vetsci-10-00216-f001:**
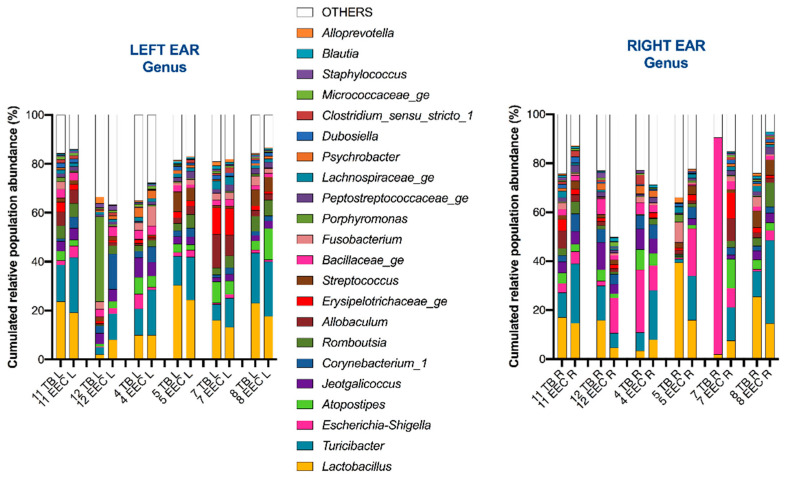
Composition of external ear canal and tympanic bulla microbiota in 12 ears from 6 healthy Beagle dogs. This bar chart shows the taxonomically annotated relative abundance of genera of all taxa detected in the left ear and right ear, collected from 6 healthy experimental Beagle dogs. Each column represents the tympanic bulla (TB) and external ear canal (EEC) microbiota of the left ear (L) or right ear (R) from healthy Beagle dogs (numbered 11, 12, 4, 5, 7 and 8).

**Figure 2 vetsci-10-00216-f002:**
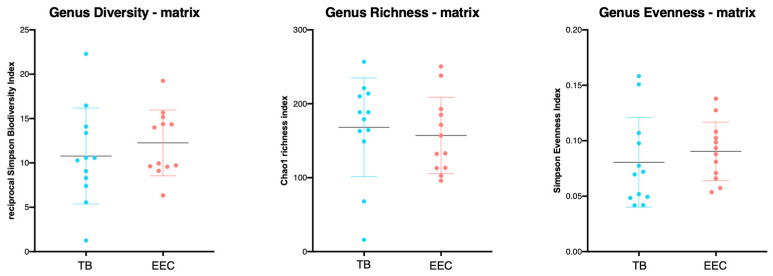
Richness, evenness and α-diversity from external ear canal and tympanic bulla in 12 ears from 6 healthy Beagle dogs. No differences of the Chao1 index (richness estimator), reciprocal Simpson index (biodiversity estimator) or the population evenness, derived from Simpson index, were observed between the EEC and the TB in experimental Beagle dogs. (TB, tympanic bulla; EEC, external ear canal).

**Figure 3 vetsci-10-00216-f003:**
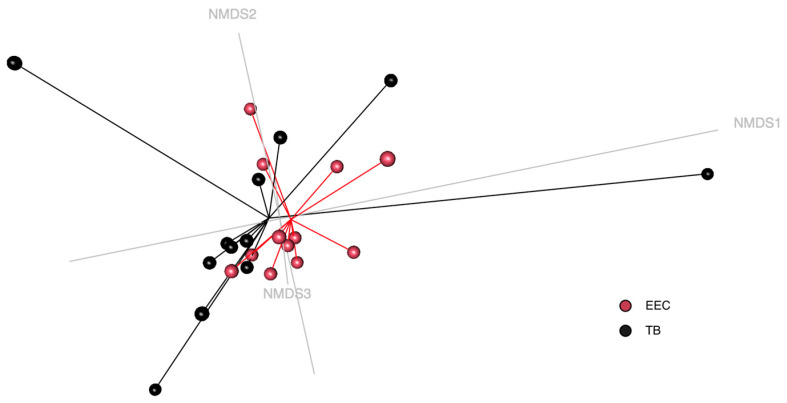
Non-metric multidimensional scaling (NMDS) ordination of external ear canal and tympanic bulla in 12 ears from 6 healthy Beagle dogs. No significant clustering was observed using AMOVA between the EEC and TB of experimental Beagle dogs. (Red dots, EEC; black dots, TB).

## Data Availability

Raw amplicon sequencing libraries were submitted to the NCBI database under bioproject number PRJNA746099.

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
