# Peer review of "The Middle Ear Microbiota in Healthy Dogs Is Similar to That of the External Ear Canal"

_vetsci, 2023, doi:10.3390/vetsci10030216_

Round 1
Reviewer 1 Report
The study is by no doubt novel and interesting. Results are significantly important to the community of veterinary dermatologists and are well presented.
Anyway, some minor rearrangements of the discussion are suggested with some additional explanations.
I would suggest to start the discussion with the paragraph from 225-234. At the end of the paragraph, before the final statement, an explanation on exclusively descendent microbial infection (from EEC to TB through damaged tympanic membrane) in dogs would be of benefit to the readers, not being specialised in dermatology. Thus, the last sentence "These bacteria were generally not observed in the tympanic bulla of healthy experimental Beagle dogs in our study" would be logical, since enterobacteria are not expected to be residents in TB.
Than the paragraph from 211-224 could be written.
Than the paragraph from 235-237 could be written. A short explanation of "occasional tympanic permeability" would be of interest with a reference cited. A significant difference between microbiota of left and right side (Figure 2) anyway should be discussed and may be here.
Line 244-245 "The high abondance of the genus Lactobacillus spp. and Turicibacter spp. "in EEC" observed in the present study..." "in EEC" should be added for better understanding.
Lines 277-279 "The present study shows a microbiota profile similar in the EEC and in the TB in healthy experimental Beagle dogs" suggestion of the conclusion: "Although the microbiota profiles of the EEC and the TB in healthy dogs could be dependent on environmental factors, the presented data are important for further studies on TB microbiota involved in OM ."
Author Response
Dear reviewer,
The authors would like to thank you for your constructive comments giving us the opportunity to improve our manuscript.
Find bellow our proposals:
- I would suggest to start the discussion with the paragraph from 225-234. At the end of the paragraph, before the final statement, an explanation on exclusively descendent microbial infection (from EEC to TB through damaged tympanic membrane) in dogs would be of benefit to the readers, not being specialised in dermatology. Thus, the last sentence "These bacteria were generally not observed in the tympanic bulla of healthy experimental Beagle dogs in our study" would be logical, since enterobacteria are not expected to be residents in TB.
- Than the paragraph from 211-224 could be written.
- Than the paragraph from 235-237 could be written.
Thank you for the suggestion, we modified the position of the paragraph as you proposed.
- A short explanation of "occasional tympanic permeability" would be of interest with a reference cited.
We added a short explanation lines 240-242: “Indeed, for unknown reasons, the eardrum could become transiently permeable. This would allow bacteria present in the EEC to migrate to the TB and seed it. This could explain the similarity of bacterial populations between the two compartments. »
- A significant difference between microbiota of left and right side (Figure 2) anyway should be discussed and may be here.
We added some discussion lines 242-271: “A significant difference between the right and the left side was observed. This observation could be explained by the hypothesis that some environmental factors could be taken into account in this process: the air flow in the kennel, the layout of the kennel or the dog’s habits for example. These environmental factors together with potential traumatic factors/mechanical insult could also explain why, in clinical settings, unilateral otitis or more severe infection on one side compared to the other are observed in some dogs. »
- Line 244-245 "The high abondance of the genus Lactobacillus spp. and Turicibacter spp. "in EEC" observed in the present study..." "in EEC" should be added for better understanding.
We added “in EEC” in the text (line 279).
- Lines 277-279 "The present study shows a microbiota profile similar in the EEC and in the TB in healthy experimental Beagle dogs" suggestion of the conclusion: "Although the microbiota profiles of the EEC and the TB in healthy dogs could be dependent on environmental factors, the presented data are important for further studies on TB microbiota involved in OM ."
Thank you for your proposal, we agree with you and modified the conclusion as you proposed from 313-315: “Although the microbiota profiles of the EEC and the TB in healthy dogs could be dependent on environmental factors, the presented data are important for further studies on TB microbiota involved in OM”
Reviewer 2 Report
they selected healthy dogs, avoiding cases of non-clinical ear problems, dogs with presence of bacteria and/or inflammatory cells at direct cytology and/or with a positive aerobic bacterial culture were excluded from this study. what do they mean non clinical ear problems are they referring to the rest of the sentence? Also I am confused as to why and importantly how they got their dogs. Ears are not sterile so it would be very hard to get samples of ears that don't have bacteria on cytology and/or culture so when they selected this group of dogs this is a biased population. What would more normal dogs look like when the compared the EEC and the middle ear? in a couple other studies one study reported that almost 50% of healthy dogs showed a positive bacterial culture. (Aoki-Komori S, Shimada K, Tani K et al. Microbial flora in the ears of healthy experimental beagles. Exp Anim 2007; 56: 67–69.) Another study reported that culture most commonly reveals coagulase-negative staphylococci, S. pseudintermedius or Bacillus spp (Lyskova P, Vydrzalova M, Mazurova J. Identification and Antimicrobial Susceptibility of Bacteria and Yeasts Isolated from Healthy Dogs and Dogs with Otitis Externa. J Vet Med A Physiol Pathol Clin Med 2007; 54: 559–563)Author Response
Dear reviewer,
The authors would like to thank you for your constructive comments giving us the opportunity to improve our manuscript.
Find below our proposals and comments:
- They selected healthy dogs, avoiding cases of non-clinical ear problems, dogs with presence of bacteria and/or inflammatory cells at direct cytology and/or with a positive aerobic bacterial culture were excluded from this study. what do they mean non clinical ear problems are they referring to the rest of the sentence?
For a better understanding, we added in the text the meaning of non-clinical ear problems from 90-93: “Indeed, some dogs may have ear infection with secondary bacterial/fungal proliferation without a history of ear problems (aural pruritus, head shaking, discharge and/or rubbing of the ear for example) mentioned by the owners or with mild signs that can be missed.”
- Also I am confused as to why and importantly how they got their dogs. Ears are not sterile so it would be very hard to get samples of ears that don't have bacteria on cytology and/or culture so when they selected this group of dogs this is a biased population. What would more normal dogs look like when the compared the EEC and the middle ear? in a couple other studies one study reported that almost 50% of healthy dogs showed a positive bacterial culture. (Aoki-Komori S, Shimada K, Tani K et al. Microbial flora in the ears of healthy experimental beagles. Exp Anim 2007; 56: 67–69.) Another study reported that culture most commonly reveals coagulase-negative staphylococci, S. pseudintermedius or Bacillus spp (Lyskova P, Vydrzalova M, Mazurova J. Identification and Antimicrobial Susceptibility of Bacteria and Yeasts Isolated from Healthy Dogs and Dogs with Otitis Externa. J Vet Med A Physiol Pathol Clin Med 2007; 54: 559–563)
The reason for this is that in order to avoid a bias of mild asymptomatic otitis externa, we preferred to exclude dogs with positive cytology and culture. We agree with you that in some dogs, the EEC samplings from healthy dogs (without ear problems) may show positive cytology/culture as mentioned in the paper you cited, but we also noticed that in the paper 50% of cases also had a negative culture in the ear canal.
Regarding cytology, our previous personal researches show a low rate of positive cytology in EEC samples from asymptomatic/healthy dogs. For the second paper you mentioned, no information was available concerning the definition of healthy dogs (no video-otoscopy information, no otis rating datas) as we know that in some cases mild signs of otitis can be missed.
To be more confident with the selection of asymptomatic healthy dogs, we decided to select those that met these criteria: negative culture and cytology.
Reviewer 3 Report
The article is very interesting and well-structured. The main limitation is the time at which samples were obtained (within 6 to 12 hours 253 after death); as discussed in the results by the authors. Another bias factor is the lack of comparison of the microbiome in the auditory tube and pharyngeal region of the same dogs. This is to understand whether, under conditions of healthy dogs, the microbial populations of the external auditory canals are similar to those of the middle ear or not.
Finally, the pathogenesis of otitis media as a descending evolution of chronic otitis externa is well documented, while it would be very useful to carry out studies on primary otitis media and in brachycephalic breeds, where often the external ear canal is not involved.
Author Response
Dear reviewer,
The authors would like to thank you for your constructive comments giving us the opportunity to improve our manuscript.
For ethical reason, it is very difficult or impossible to select dogs from different breeds. We cannot perform samples of the tympanic bulla under aseptic condition without a surgery (TECALBO). However, it was not the objective of this study to include different breeds (especially brachycephalic), but it would have been very interesting to have these data to compare the different type of breeds. Actually, we are presently comparing the microbiota in the EEC and the TB in dogs with chronic suppurative otitis that need a TECALBO surgery (in different breeds including brachycephalic dogs): results will be available in a near future.